# Unsupervised Domain Adaptation for Abdominal Organ Segmentation Using Pseudo Labels and Organ Attention CycleGAN

Jianghao Wu[1][0000−0001−9743−9316], Guoning Zhang[3][0009−0007−4186−3290], Xiaoran Qi[4], Huamin Wang[1][0009−0001−7872−6499], Xinya Liu[1], and Guotai Wang[1,2*][0000−0002−8632−158X]

[1] School of Mechanical and Electrical Engineering, University of Electronic Science and Technology of China, Chengdu, 611731, China
`guotai.wang@uestc.edu.cn`
[2] Shanghai Artificial Intelligence Laboratory, Shanghai, 200030, China
[3] School of Information Communication Engineering, University of Electronic Science and Technology of China, Chengdu, 611731, China
[4] School of Automation Engineering, University of Electronic Science and Technology of China, Chengdu, 611731, China

**Abstract.** Abdominal organ segmentation in MRI scans poses significant challenges due to the scarcity of annotated data and the substantial domain shift between MRI and more readily available CT scans. In response to these challenges, we propose a novel approach leveraging Organ Attention CycleGAN for unsupervised domain adaptation (UDA) in abdominal organ segmentation. Our method begins by translating labeled CT images into corresponding MRI modalities using an enhanced CycleGAN model that incorporates an organ attention mechanism. This mechanism ensures the preservation of critical anatomical structures during the translation process. Following the image translation, we employ the nnU-Net V2 framework, enhanced with Residual Encoder Presets, to perform fully supervised segmentation training on the translated MRI images. This combination allows our model to leverage the extensive labeled CT datasets effectively and adapt them to the MRI domain, achieving robust segmentation performance without requiring annotated MRI data. To further refine the model's performance, we introduce a self-training process using a prediction consistency algorithm. By generating multiple predictions via 5-fold cross-validation and evaluating their consistency using the Dice coefficient, we select the most reliable pseudo labels for additional training. This approach enables our model to improve segmentation accuracy on real MRI scans. Our method was evaluated on the official validation set of the MICCAI FLARE 2024 TASK3, achieving promising results with an Organ DSC of 0.77 and an Organ NSD of 0.83, further highlighting the effectiveness of our approach in addressing the challenges of UDA for abdominal organ segmentation.

**Keywords:** Unsupervised domain adaptation · abdominal organ segmentation · organ attention cycleGAN

## 1   Introduction

Abdominal organ segmentation in medical images is essential for various clinical applications, including computer-aided diagnosis, surgical planning, and radiotherapy [36]. Significant advancements have been made in this field over the past decade, particularly with the use of convolutional neural networks (CNNs)[22,13]. However, most progress has focused on segmentation in computed tomography (CT) images. While magnetic resonance imaging (MRI) is gaining attention due to its superior soft tissue contrast and the absence of ionizing radiation, abdominal organ segmentation in MRI remains under-explored. This is primarily due to the scarcity of annotated MRI scans, which presents a significant challenge for training fully supervised models[5].

The shortage of annotated MRI datasets stands in stark contrast to the abundance of labeled CT scans [3]. For example, the MICCAI AMOS challenge includes only 40 labeled MRI scans in its training set, while annotated CT datasets are far more available [12]. This disparity raises a crucial question: how can effective abdominal MRI segmentation models be developed without relying on MRI annotations [18]? This challenge is particularly important given MRI's clinical relevance in diagnosing and treating abdominal diseases [24]. Addressing this question is challenging, as models trained on CT data often perform poorly on MRI datasets due to the significant domain gap between CT and MRI, where their data distributions differ substantially [32].

Unsupervised domain adaptation (UDA) has shown promise in addressing domain shifts in medical image segmentation by transferring knowledge from labeled CT scans (source domain) to unlabeled MRI scans (target domain) without requiring MRI annotations [33,31]. UDA methods typically align source and target domains through image appearance, feature distribution, or output structure [26,14]. Image translation techniques, such as CycleGAN [39] and Contrastive Unpaired Translation (CUT)[23], align image appearances but often distort anatomical structures, reducing segmentation accuracy[34]. To improve alignment, Dou et al.[4] introduced adversarial loss to align feature spaces, while Wu et al.[30] used characteristic function distance to reduce distribution discrepancies. Output alignment methods further ensure structural consistency between predictions in the two domains, which is essential for medical segmentation where anatomical structures vary [27]. Despite progress, most UDA methods focus on 2D images and struggle with 3D medical segmentation. DAR-Net [38] combines 2D style transfer with 3D segmentation but still faces domain gaps and limited performance due to unrealistic style transfer and insufficient training data.

In this work, we propose a novel approach leveraging the Organ Attention CycleGAN to tackle the challenges of unsupervised domain adaptation (UDA) for abdominal organ segmentation in MRI scans. Our method consists of three key stages: First, we preprocess annotated CT images by extracting 2D slices and categorizing them into eight distinct MRI modalities, including DWI, T2WI, contrast-enhanced (C+A), InPhase, OutPhase, C+pre, C+V, and C+Delay. These modalities are derived from the LLD-MMRI dataset [15], with 50 samples from each modality used for image generation and conversion. Next, we

employ CycleGAN, enhanced with an organ attention mechanism, to translate the CT images into multi-modal MRI-like images, ensuring the preservation of anatomical structures and improving segmentation accuracy. These translated images are then used to train a nnU-Net V2 segmentation network, incorporating Residual Encoder Presets for robust organ segmentation in the MRI domain. Finally, inspired by [35], we introduce a self-training process where the trained nnU-Net models generate predictions on real MRI scans. A novel prediction consistency algorithm refines these predictions by selecting high-quality pseudo labels for further training, ultimately enhancing segmentation performance. This approach effectively utilizes the rich, labeled CT datasets to adapt to the MRI domain, overcoming the scarcity of MRI annotations and achieving significant improvements in multi-organ segmentation without relying on MRI labels.

The main contributions of this work are as follows:

- We propose a novel Organ Attention CycleGAN framework for cross-modality image translation, which effectively preserves anatomical structures during the translation from CT to multiple MRI modalities.
- We demonstrate the effectiveness of using a carefully curated subset of MRI modalities from the LLD-MMRI dataset, highlighting the importance of modality-specific training in UDA for medical image segmentation.
- We introduce a prediction consistency algorithm for selecting high-quality pseudo labels during the self-training process, leading to significant improvements in segmentation accuracy on real MRI data.

## 2    Method

In this section, we detail our proposed approach for unsupervised domain adaptation (UDA) in abdominal organ segmentation using the Organ Attention CycleGAN. As shown in Fig. 1, our method is designed to bridge the gap between labeled CT images and unlabeled MRI images by leveraging a novel image translation and segmentation framework. The process is divided into three key stages: (1) Data preprocessing and modality selection, where annotated CT images are converted into various MRI modalities; (2) Cross-modality image translation using the Organ Attention CycleGAN, which preserves critical anatomical structures during the conversion from CT to MRI; and (3) Segmentation and self-training, where the translated images are used to train a nnU-Net V2 segmentation network, followed by a consistency-based pseudo label refinement process to further enhance the segmentation performance on real MRI scans. Each of these stages is crucial in achieving accurate and robust organ segmentation across different MRI modalities without relying on labeled MRI data.

### 2.1    Organ Attention CycleGAN

Let $\mathcal{D}_s$ and $\mathcal{D}_t$ denote a set of labeled source-domain images and a set of unlabeled target-domain images, respectively. Let $X_i^s$ and $X_j^t$ represent the $i$-th

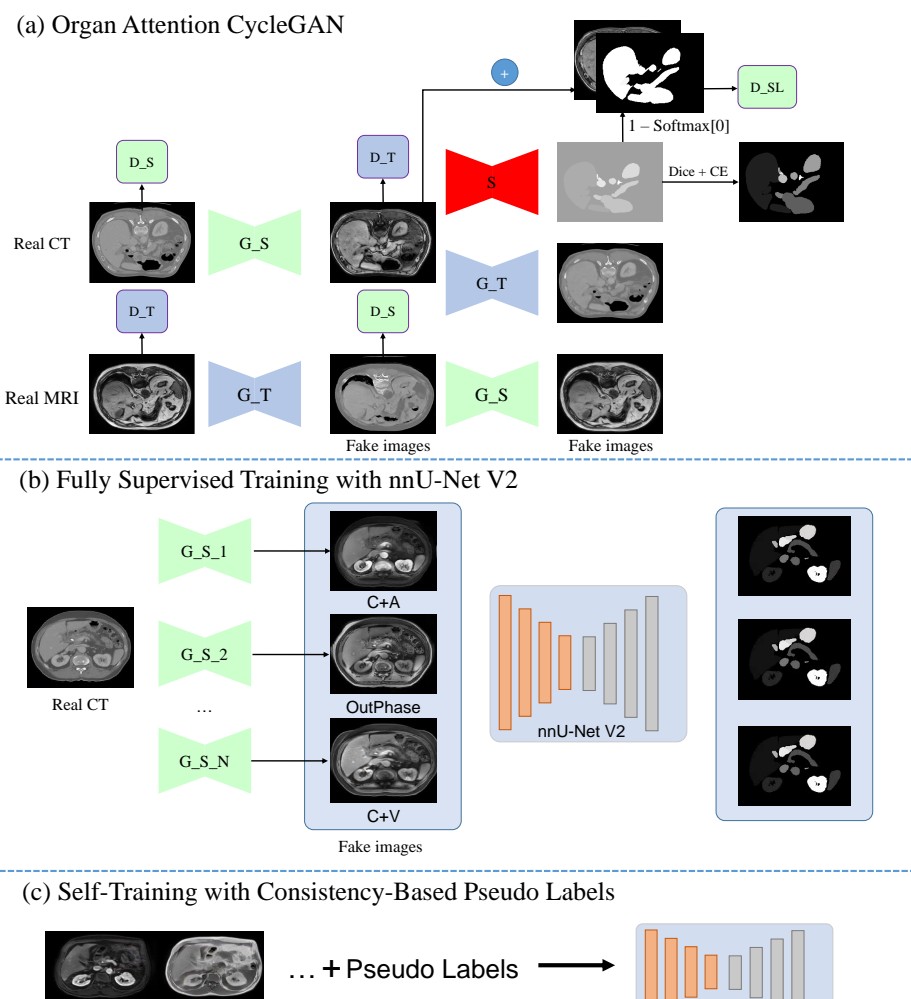

**Fig. 1.** The proposed pipeline. (a) Organ Attention CycleGAN, which enhances the accuracy of CT-to-MRI translation by focusing on organ-specific features during the image generation process. (b) Fully Supervised Training with nnU-Net V2, where synthetic MRI images and real CT labels are used to jointly train a segmentation network. (c) Self-Training with Consistency-Based Pseudo Labels, where real MRI images and their corresponding pseudo labels are used to further fine-tune the segmentation network for improved performance on real MRI data.

image from $\mathcal{D}_s$ and the $j$-th image from $\mathcal{D}_t$, where the label of $X_i^s$ is $Y_i^s$. Since the source domain and target domain images are from different patient groups, $X_i^s$ and $X_j^t$ are unpaired, i.e., they come from different patients. Due to the significant domain shift between $\mathcal{D}_s$ and $\mathcal{D}_t$, directly training a model on $\mathcal{D}_s$ and applying it to generate pseudo labels for $\mathcal{D}_t$ would likely result in poor performance [31].

To address this issue, CycleGAN [39] has been widely adopted as a solution for cross-domain image translation. CycleGAN employs two image style translators, $G_t$ and $G_s$, to translate images from the source domain to the target domain and vice versa. Specifically, given a labeled source-domain image $X_i^s$, $G_t$ translates it into a pseudo target-domain image $X_i^{s \to t} = G_t(X_i^s)$, and $G_s$ then translates $X_i^{s \to t}$ back to the source domain, producing a pseudo source-domain image $X_i^{s'} = G_s(X_i^{s \to t})$. These translators are trained jointly using unpaired datasets, optimizing for both adversarial losses $\mathcal{L}_{gan}^t$, $\mathcal{L}_{gan}^s$, and cycle consistency losses $\mathcal{L}_{cyc}$. The adversarial losses ensure that the translated images are indistinguishable from real images in the target or source domains:

$$
\begin{aligned}
\mathcal{L}_{gan}^t(G_t, D_t) =& \mathbb{E}_{X_j^t \sim \mathcal{D}_t}[\log D_t(X_j^t)] \\
&+ \mathbb{E}_{X_i^s \sim \mathcal{D}_s}[\log(1 - D_t(X_i^{s \to t}))],
\end{aligned}
\tag{1}
$$

$$
\begin{aligned}
\mathcal{L}_{gan}^s(G_s, D_s) =& \mathbb{E}_{X_i^s \sim \mathcal{D}_s}[\log D_s(X_i^s)] \\
&+ \mathbb{E}_{X_i^{s \to t} \sim \mathcal{D}_t}[\log(1 - D_s(X_i^{s'}))],
\end{aligned}
\tag{2}
$$

while the cycle consistency losses ensure that the translated images can be converted back to the original domain without significant alterations:

$$
\begin{aligned}
\mathcal{L}_{cyc}(G_s, G_t) =& \mathbb{E}_{X_i^s \sim \mathcal{D}_s}[\|G_s(G_t(X_i^s)) - X_i^s\|_1] \\
&+ \mathbb{E}_{X_j^t \sim \mathcal{D}_t}[\|G_t(G_s(X_j^t)) - X_j^t\|_1].
\end{aligned}
\tag{3}
$$

However, CycleGAN has inherent limitations when applied to medical image translation, particularly for tasks like abdominal organ segmentation. Since the training sets are unpaired, achieving an exact match between the translated images and their ground truth counterparts is challenging. This can lead to structural distortions in the translated images, such as artifacts or inaccuracies in the anatomical regions of interest. These distortions can significantly affect downstream tasks like segmentation, where precise anatomical structure is crucial. Moreover, CycleGAN primarily focuses on appearance translation without explicitly preserving the critical anatomical features necessary for accurate medical image analysis.

To overcome these limitations, we propose the Organ Attention CycleGAN, which integrates an organ attention mechanism directly into the CycleGAN framework. This mechanism is designed to focus on the regions of interest corresponding to specific organs, ensuring that these areas are highlighted and preserved during the translation from the source domain to the target domain.

Given a source domain image $X_i^s$, the generator $G_t$ first produces a translated image $X_i^{s \to t} = G_t(X_i^s)$. To focus on the critical anatomical regions during the

translation process, we incorporate an organ attention mechanism that leverages the segmentation predictions from a pre-trained segmentation network $S$.

Specifically, the segmentation network $S$ generates a segmentation prediction $S(X_i^s)$ for the translated image $X_i^{s \rightarrow t}$. This prediction includes multiple channels, each representing a different organ or background. To create an attention map that highlights the organs of interest, we apply a softmax activation function to the segmentation output, normalizing the predictions across the channels:

$$\text{att\_map}(X_i^s) = \text{softmax}(S(X_i^s)). \tag{4}$$

The first channel of this prediction typically corresponds to the background, so the organ attention map $A(X_i^s)$ is computed by subtracting the first channel from 1:

$$A(X_i^s) = 1 - \text{att\_map}(X_i^s)[:, 0, :, :], \tag{5}$$

where $\text{att\_map}(X_i^s)[:, 0, :, :]$ represents the background channel. This attention map $A(X_i^s)$ effectively highlights the regions corresponding to the organs of interest.

The attention map is then applied to the translated image $X_i^{s \rightarrow t}$ to enhance the focus on these critical regions. Depending on the chosen fusion method, we combine the attention map with the translated image in additive, the attention-enhanced image $\hat{X}_i^{s \rightarrow t}$ is computed as:

$$\hat{X}_i^{s \rightarrow t} = X_i^{s \rightarrow t} + A(X_i^s). \tag{6}$$

The attention-enhanced image $\hat{X}_i^{s \rightarrow t}$ is then passed through the local discriminator $D_t^{\text{local}}$ to ensure that it is indistinguishable from real target domain images while preserving the anatomical structures:

$$\begin{aligned}
\mathcal{L}_{gan}^t(G_t, D_t^{\text{local}}) = & \mathbb{E}_{X_j^t \sim \mathcal{D}_t}[\log D_t^{\text{local}}(X_j^t)] \\
& + \mathbb{E}_{X_i^s \sim \mathcal{D}_s}[\log(1 - D_t^{\text{local}}(\hat{X}_i^{s \rightarrow t}))].
\end{aligned} \tag{7}$$

By incorporating the organ attention mechanism, our Organ Attention CycleGAN ensures that the translated images retain critical anatomical structures, leading to more accurate and reliable segmentation results in the target domain.

### 2.2   Fully Supervised Training with nnU-Net V2

After translating the labeled CT images into corresponding MRI modalities using the Organ Attention CycleGAN, the next step involves training a segmentation network in a fully supervised manner. For this purpose, we employ the nnU-Net V2 framework, which has established itself as a state-of-the-art solution in 3D medical image segmentation. nnU-Net V2 is particularly effective because it provides a highly configurable U-Net architecture that can be adapted to a wide range of medical imaging tasks, ensuring robust performance across different datasets.

In our approach, we specifically utilize the Residual Encoder Presets within the nnU-Net framework. This choice is motivated by the findings of recent studies, which have demonstrated that many claims of superior performance by novel architectures often fail to hold when subjected to rigorous validation. Instead, employing well-established CNN-based U-Net models, such as those incorporating ResNet or ConvNeXt variants, within the nnU-Net framework has proven to yield state-of-the-art results. Furthermore, scaling these models to modern hardware resources further enhances their performance, making them a reliable choice for our segmentation task.

The translated MRI images from the 8 sequences, including DWI, T2WI, C+A, InPhase, OutPhase, C+pre, C+V, and C+Delay, are used as input to train the nnU-Net V2 model. During training, the model learns to accurately segment multiple abdominal organs by leveraging the rich and diverse information provided by these different MRI modalities. By using the Residual Encoder Presets, the model benefits from improved feature representation and network depth, allowing it to capture complex anatomical structures with greater precision.

Overall, the integration of the Organ Attention CycleGAN for image translation and the nnU-Net V2 for segmentation provides a powerful framework for achieving high-quality abdominal organ segmentation in MRI, even in the absence of annotated MRI data. This approach not only addresses the challenge of domain adaptation but also ensures that the segmentation network is trained in a fully supervised manner, leveraging the strengths of both advanced image translation and robust segmentation methodologies.

### 2.3   Self-Training with Consistency-Based Pseudo Labels

In the third stage of our approach, we implement a self-training process to further refine the segmentation performance on real MRI images. After training the nnU-Net V2 model on the translated MRI images, we perform inference on all real MRI scans using a 5-fold cross-validation strategy. This process generates five different segmentation predictions for each MRI scan.

To filter out noisy predictions and improve the overall segmentation accuracy, we introduce a prediction consistency algorithm. This algorithm evaluates the consistency of each prediction by calculating the Dice coefficient between pairs of predictions across the five folds. For each organ class, we compute the average consistency score, which serves as a measure of the reliability of the predictions for that class.

The consistency score for each organ class is computed as follows:

$$\text{Consistency}(c) = \frac{1}{\binom{N}{2}} \sum_{i=1}^{N-1} \sum_{j=i+1}^{N} \text{Dice}(P_i^c, P_j^c) \tag{8}$$

where $N$ is the number of models (in our case, $N = 5$), $c$ represents the organ class, $P_i^c$ and $P_j^c$ are the predictions for class $c$ from models $i$ and $j$, $\text{Dice}(P_i^c, P_j^c)$ is the Dice coefficient between predictions $P_i^c$ and $P_j^c$ for class $c$.

After calculating the consistency scores for each organ class, we rank the real MRI samples based on these scores. For each MRI modality, we select the top 50 samples with the highest consistency scores and use their predictions as pseudo labels for further training.

To generate the final pseudo labels, we average the five predictions for each pixel across the folds and assign the most consistent label to each pixel. The final pseudo label for each pixel is determined as follows:

$$\text{Final Label}(x, y) = \arg\max_c \left( \frac{1}{N} \sum_{i=1}^{N} P_i^c(x, y) \right) \tag{9}$$

where $(x, y)$ denotes the coordinates of the pixel, $P_i^c(x, y)$ is the probability that pixel $(x, y)$ belongs to class $c$ in the $i$-th model's prediction, $\arg\max_c$ selects the class $c$ with the highest average probability.

By utilizing the most consistent predictions as pseudo labels, we perform additional supervised training on these selected samples, effectively refining the model's segmentation performance on real MRI scans. This self-training process leverages the strength of consensus among predictions to improve the robustness and accuracy of the segmentation network, particularly in challenging cases where annotated MRI data is not available.

### 2.4   Resource usage and inference speed

In our approach, we utilized unannotated MRI images and generated high-quality pseudo labels for the next stage of training.

However, for the core training process, we relied exclusively on real annotated CT data and did not employ pseudo labels generated by other methods. Specifically, we did not incorporate the pseudo labels produced by the FLARE22 winning algorithm [10] or the best-accuracy algorithm [28]. Our method is centered on leveraging the true CT annotations for unsupervised domain adaptation to MRI, ensuring that the segmentation results are purely based on real labeled data, without the use of external pseudo label generation techniques.

To accelerate inference speed, we transferred the data preprocessing operations to the GPU, which provides a significant boost in processing time compared to CPU-based operations. This allows for more efficient handling of data during inference. Furthermore, to minimize resource consumption, we adopted models of standard sizes, ensuring optimal memory usage while maintaining strong performance. These optimizations are fully integrated within the nnU-Net V2 framework, which facilitates both faster inference and resource-efficient operation.

## 3   Experiments

### 3.1   Dataset and Evaluation Measures

The dataset for the challenge is curated from over 30 medical centers, licensed for research use, and includes a combination of CT and MRI scans from well-

known datasets such as TCIA [2], LiTS [1], MSD [25], KiTS [8,9], autoPET [7,6], AMOS [11], LLD-MMRI [16], TotalSegmentator [29], AbdomenCT-1K [21], and past FLARE Challenges [17,19,20]. This broad collection provides extensive data for training, validation, and testing across various medical centers and modalities.

For Task 3 of this challenge, we specifically focused on the following datasets:

**CT Scans:** We used 2,300 CT scans, of which 50 cases are labeled with ground-truth annotations. The remaining CT cases were annotated using pseudo labels generated by the FLARE22 winning algorithm [10], achieving an approximate Dice score (DSC) of 90%.

**MRI Scans:** The MRI dataset consists of unlabeled scans from two primary sources:

The AMOS dataset, which provides a significant portion of the MRI scans. The LLD-MMRI dataset, which includes MRI images across 8 different modalities (DWI, T2WI, C+A, InPhase, OutPhase, C+pre, C+V, and C+Delay), enhancing the diversity of contrasts and anatomical information used for training. The segmentation task focuses on identifying and segmenting 13 organ classes, such as the liver, kidneys, spleen, and pancreas.

**Validation and Testing Sets:** The validation set includes 110 MRI scans, with 60 cases from the AMOS dataset and 50 from U-Mamba experiments. The testing set comprises 300 MRI scans, covering various MRI sequences from centers that were not included in the training or validation sets.

**Evaluation Measures:** We use Dice Similarity Coefficient (DSC) and Normalized Surface Dice (NSD) to assess accuracy, alongside running time and area under the GPU memory-time curve for efficiency. The tolerance for running time is 60 seconds.

### 3.2   Implementation details

**Pre-processing:** Before training the models, we applied a series of preprocessing steps to the 3D medical images. For CT images, we performed clipping using a window of [-600, 600], while for MRI images, we clipped the intensity values between the 1st and 99th percentiles. Following this, all images were linearly normalized to a range of [-1, 1].

**GAN Training:** To train our Organ Attention CycleGAN, we first converted the 3D images into 2D slices by slicing along the Z-axis. At this stage, all 2D slices were retained without any filtering. We trained the CycleGAN for 100 epochs, with each epoch consisting of 2,400 iterations. Due to GPU memory constraints, the batch size was set to 2.

Upon completion of the training, we used the trained CycleGAN to translate the CT images into corresponding MRI modalities. These translated images were then reassembled into 3D volumes based on the original slice information. Given the large volume of MRI samples, we randomly selected 50 MRI samples to train the Organ Attention CycleGAN, ensuring computational feasibility.

We adopted the standard CycleGAN architecture, where both the generators and discriminators follow the original ResNet-based encoder-decoder design [40].

The organ attention mechanism was integrated into this baseline without altering the core generator or discriminator structures. This design choice ensures compatibility with existing training pipelines and maintains architectural simplicity.

**nnU-Net V2 Training:** For the nnU-Net V2 framework, as shown in Table 1, we adhered strictly to the default pipeline and dataset partitioning provided by nnU-Net, specifically utilizing the 'nnUNetResEncUNetMPlans' configuration. The 3D full-resolution configuration was used with a data identifier 'nnUNetPlans 3d fullres' and a default preprocessor ('DefaultPreprocessor'). The batch size was limited to 2 due to GPU memory constraints. Each input image was preprocessed into patches of size [48, 160, 224], with a median voxel size of [72.0, 232.0, 292.0] and a spatial resolution of [3.0, 1.3, 1.3] mm. CT images were normalized using a 'CTNormalization' scheme. No manual modifications were made to the planning or dataset configuration. We apply data augmentation on the fly during training, including additive brightness, gamma correction, rotation, scaling, elastic deformation, and non-linear transformations.

**Table 1.** Training protocols.

| | |
|---|---|
| Network initialization | "he" normal initialization |
| Batch size | 2 |
| Patch size | $48 \times 160 \times 224$ |
| Total epochs | 200 |
| Optimizer | SGD with nesterov momentum ($\mu = 0.99$) |
| Initial learning rate (lr) | 0.01 |
| Lr decay schedule | Poly learning rate policy: $(1 - \text{epoch}/200)^{0.9}$ |
| Training time | 8 hours |
| Loss function | Dice loss and cross entropy loss |
| Number of model parameters | 88.02M |
| Number of flops | 187G |

Initially, we combined the CT-to-MRI translated images across the 8 MRI modalities with their corresponding CT labels to train a nnU-Net model. This model was trained using a 5-fold cross-validation strategy, with each fold undergoing 200 epochs of training.

After completing the training, we used the model from the final epoch of the cross-validation to perform inference on the real MRI scans. From these inferences, we selected the top 50 MRI samples with the highest consistency in their predictions. These high-consistency predictions were then used as pseudo labels for a subsequent round of training. The final segmentation network was trained once more using 5-fold cross-validation, each fold again trained for 200 epochs.

**Post-processing:** We implemented a post-processing pipeline for MRI image segmentation to improve the accuracy of the segmented structures. The process involves identifying and adjusting the bounding boxes of segmented regions, applying organ-specific rules such as filling small holes, and retaining only the largest connected components to reduce noise. Additionally, we remove regions below a certain volume threshold for organs like the gallbladder and spleen, ensuring that only the most relevant anatomical structures are preserved.

**Environment settings:** The development environments and requirements are presented in Table 2.

**Table 2.** Development environments and requirements.

| Operating System | Ubuntu 20.04.3 LTS |
|---|---|
| CPU | Intel(R) Xeon(R) CPU E5-2678 v3 @ 2.50GHz |
| RAM | 125GiB |
| GPU | NVIDIA GeForce RTX 2080 Ti (11GB) |
| CUDA version | 12.5 |
| Programming language | Python 3.12.4 |
| Deep learning framework | PyTorch 2.4.0+cu121 |
| Code | https://github.com/JianghaoWu/FLARE24-Task3 |

## 4    Results

### 4.1    Quantitative results on validation set

Table 4 presents the Dice Similarity Coefficient (DSC) and Normalized Surface Dice (NSD) values for 13 organs on the validation dataset. Segmentation performance varies across organs, with the highest DSC values for the liver (0.9426), right kidney (0.9345), and spleen (0.9395), reflecting their simpler structures and strong representation in the training data.

Lower performance is observed for challenging structures like the duodenum (DSC 0.5752) and right adrenal gland (DSC 0.5811) due to their smaller, irregular shapes and complex anatomical locations. NSD values follow a similar trend, with the spleen achieving 0.9584, while the inferior vena cava (IVC) and gallbladder have lower scores of 0.7077 and 0.6242, respectively. The overall average DSC is 0.7674, with an NSD of 0.8319, highlighting solid performance but leaving room for improvement in smaller, more complex organs.

Unlabelled MRI cases with pseudo-labeling improved segmentation, particularly for the liver and kidneys, where pseudo labels offered strong guidance. However, this effect was less pronounced for smaller, less distinct structures like the duodenum due to labeling ambiguities.

The method performs well on larger organs with clear boundaries, such as the liver, right kidney, and spleen, which are less affected by image artifacts or adjacent structures. In contrast, segmentation struggles with smaller, complex

organs like the adrenal glands and duodenum, likely due to shape variability and limited high-quality training data.

In this ablation study, stage 1 results are obtained by training the model on MRI images generated from real CT scans. As shown in Table 5, the model achieves a DSC of 92.87% and an NSD of 91.87% for the liver, as well as a DSC of 86.17% and an NSD of 86.08% for the right kidney.

In stage 2, the model is further trained with pseudo-labels generated from MRI data. This second stage shows an improvement in most organs, with the liver achieving a DSC of 94.26% and an NSD of 83.96%, and the right kidney showing a DSC of 93.45% and an NSD of 93.53%. The overall average performance also improves from 68.49% to 76.74% in DSC and from 74.22% to 83.19% in NSD, demonstrating the effectiveness of using pseudo-labels for further training.

All images complete preprocessing, inference, and post-processing within one minute, with GPU memory usage for inference kept under 10GB per image, as shown in Table 3.

**Table 3.** Quantitative evaluation of segmentation efficiency in terms of the running time and GPU memory consumption.

| Case ID | Image Size | Fold-level Running Time (s) | Max GPU (MB) |
|---|---|---|---|
| amos 0588 | (512, 512, 168) | 5.20 | 2003 |
| amos 0507 | (320, 290, 72) | 5.50 | 1921 |
| amos 7789 | (1024, 2024, 32) | 4.01 | 1799 |

**Table 4.** Quantitative evaluation of segmentation performance on the validation dataset.

| Target | Validation | |
|---|---|---|
| | DSC(%) | NSD(%) |
| Liver | 94.26 | 83.96 |
| Right kidney | 93.45 | 93.53 |
| Spleen | 93.95 | 95.84 |
| Pancreas | 76.55 | 88.22 |
| Aorta | 87.16 | 90.99 |
| Inferior vena cava | 68.98 | 70.77 |
| Right adrenal gland | 58.11 | 75.48 |
| Left adrenal gland | 65.60 | 79.92 |
| Gallbladder | 65.29 | 62.42 |
| Esophagus | 64.62 | 80.82 |
| Stomach | 80.21 | 84.79 |
| Duodenum | 57.52 | 80.89 |
| Left kidney | 91.97 | 93.82 |
| Average | 76.74 | 83.19 |

**Table 5.** Quantitative evaluation results of ablation on stage 1 and 2.

| Target | First stage | | Second stage | |
|---|---|---|---|---|
| | DSC(%) | NSD(%) | DSC(%) | NSD (%) |
| Liver | 92.87 | 91.87 | 94.26 | 83.96 |
| Right kidney | 86.17 | 86.08 | 93.45 | 93.53 |
| Spleen | 85.33 | 85.44 | 93.95 | 95.84 |
| Pancreas | 68.02 | 79.78 | 76.55 | 88.22 |
| Aorta | 78.93 | 81.03 | 87.16 | 90.99 |
| Inferior vena cava | 64.13 | 64.88 | 68.98 | 70.77 |
| Right adrenal gland | 55.53 | 71.22 | 58.11 | 75.48 |
| Left adrenal gland | 57.03 | 68.29 | 65.60 | 79.92 |
| Gallbladder | 52.20 | 49.54 | 65.29 | 62.42 |
| Esophagus | 49.66 | 60.42 | 64.62 | 80.82 |
| Stomach | 65.32 | 69.97 | 80.21 | 84.79 |
| Duodenum | 47.64 | 67.72 | 57.52 | 80.89 |
| Left kidney | 87.51 | 88.58 | 91.97 | 93.82 |
| Average | 68.49 | 74.22 | 76.74 | 83.19 |

### 4.2    Qualitative results

The qualitative results shown in Fig. 2 display four representative segmentation outcomes from our model. In column (a), the original MRI images are presented, while column (b) shows the ground truth annotations. Columns (c) and (d) illustrate the segmentation results from the first stage and second stage, respectively. The results from the second stage (d), which incorporate self-training with consistency-based pseudo labels, demonstrate improved segmentation performance, particularly in the more complex anatomical regions, compared to the first stage (c).

As shown in Fig. 3, the CT images are translated into eight different MRI modalities, each providing unique contrast and anatomical details. These modalities include DWI, T2WI, C+A, InPhase, OutPhase, C+pre, C+V, and C+Delay. The variety of MRI sequences enables the model to capture diverse tissue characteristics and improves the robustness of the segmentation across different anatomical structures.

### 4.3    Results on final testing set

As shown in Tab. 6, our method achieved a mean Dice Similarity Coefficient (DSC) of 65.7% and a mean Normalized Surface Dice (NSD) of 68.7% on the final testing set. The median DSC and NSD further indicate consistent performance, with values of 73.6% and 79.1%, respectively. The average inference time per case was 64.1 seconds, demonstrating practical efficiency. In terms of computational resources, the average GPU memory usage was approximately 4.1 million bytes, reflecting a lightweight model suitable for deployment.

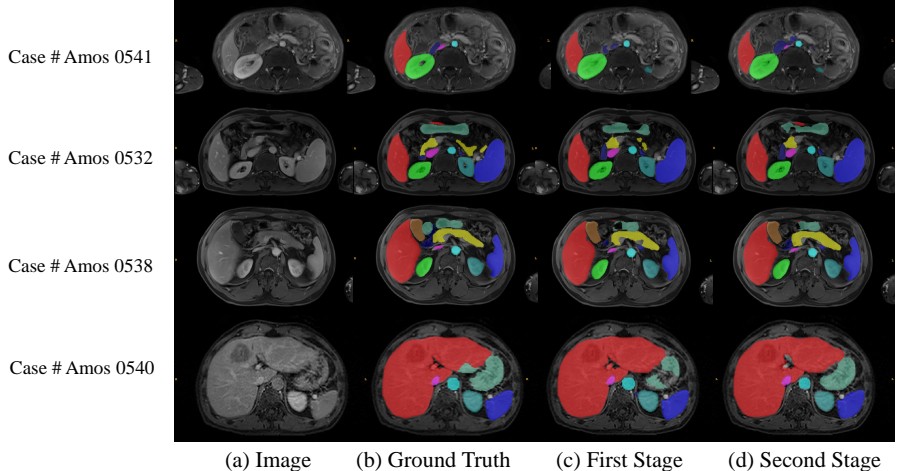

Case # Amos 0541

Case # Amos 0532

Case # Amos 0538

Case # Amos 0540

(a) Image        (b) Ground Truth        (c) First Stage        (d) Second Stage

**Fig. 2.** Qualitative results of our nnU-Net model. (c) shows the results from the first stage: fully supervised training with nnU-Net V2 using synthetic MRI images and real CT labels. (d) presents the results from the second stage: self-training with consistency-based pseudo labels generated from real MRI images.

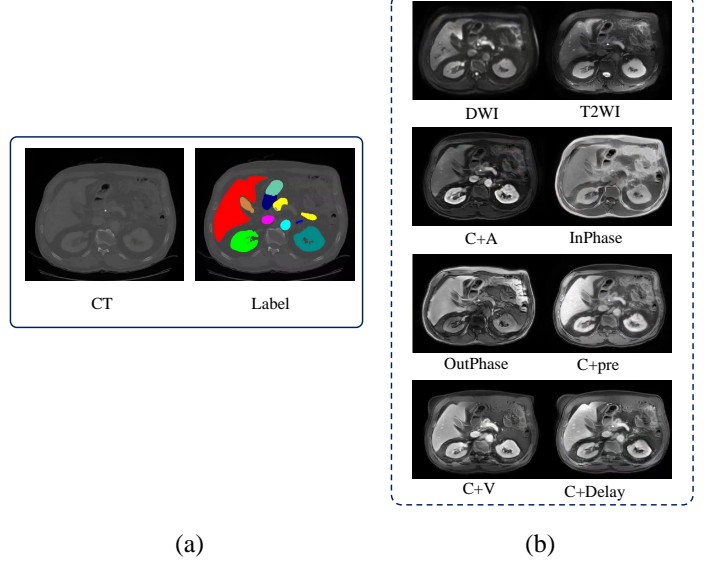

(a)                            (b)

**Fig. 3.** Qualitative results of image translation. (a) CT image and its corresponding labeled segmentation. (b) The same CT image translated into eight MRI-like modalities—DWI, T2WI, C+A, InPhase, OutPhase, C+pre, C+V, and C+Delay—using our Organ Attention CycleGAN. These synthetic images demonstrate the model's ability to preserve anatomical structure across diverse MRI contrasts under an unpaired setting.

**Table 6.** Quantitative evaluation results on the test set.

| Metric | Ours |
| --- | --- |
| DSC mean | 65.7 ± 20.1 |
| DSC median | 73.6 (50.1, 83.2) |
| NSD mean | 68.7 ± 23.2 |
| NSD median | 79.1 (49.8, 89.2) |
| Time mean (s) | 64.1 ± 30.4 |
| Time median (s) | 55.7 (47.2, 70.8) |
| GPU mean (bytes) | 4,127,758.4 ± 2,106,414.8 |
| GPU median (bytes) | 3,516,795.2 (2,931,383.9, 4,624,098.1) |

### 4.4   Limitation and Future Work

Our core approach primarily focuses on image generation and conversion, as well as the selection and filtering of pseudo labels, without introducing any improvements to the fully supervised methods. Currently, we have relied solely on nnU-Net V2 as our primary framework. Additionally, the current approach involves training separate CycleGAN models for each MRI modality to preserve modality-specific anatomical features. While effective, this design introduces additional computational cost and may limit scalability. Future work will explore unified or multi-modal image translation frameworks to reduce redundancy and improve efficiency. Moreover, although our method demonstrates strong performance within the FLARE24 framework, we did not include comparisons with existing UDA baselines such as AdaptSegNet or SIFA. Incorporating such benchmarks will be a key focus in future extensions to further validate the effectiveness and generalizability of our approach.

In the future, we plan to explore a range of fully supervised enhancements specifically tailored for multi-modal MRI, aiming to achieve further advancements in segmentation performance.

**Acknowledgements**  The authors of this paper declare that the segmentation method they implemented for participation in the FLARE 2024 challenge has not used any pre-trained models nor additional datasets other than those provided by the organizers. The proposed solution is fully automatic without any manual intervention. We thank all data owners for making the CT scans publicly available and CodaLab [37] for hosting the challenge platform.

## Disclosure of Interests

The authors declare no competing interests.

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

**Table 7.** Checklist Table. Please fill out this checklist table in the answer column.

| Requirements | Answer |
| --- | --- |
| A meaningful title | Yes |
| The number of authors ($\leq 6$) | 6 |
| Author affiliations and ORCID | Yes |
| Corresponding author email is presented | Yes |
| Validation scores are presented in the abstract | Yes |
| Introduction includes at least three parts: background, related work, and motivation | Yes |
| A pipeline/network figure is provided | Figure 1 |
| Pre-processing | Page 9 |
| Strategies to use the partial label | Page 8 |
| Strategies to use the unlabeled images. | Page 7, 8 |
| Strategies to improve model inference | Page 8 |
| Post-processing | Page 10 |
| The dataset and evaluation metric section are presented | Page 8, 9 |
| Environment setting table is provided | Table 2 |
| Training protocol table is provided | Table 1 |
| Ablation study | Page 11 |
| Efficiency evaluation results are provided | Table 12 |
| Visualized segmentation example is provided | Figure 13 |
| Limitation and future work are presented | Yes |
| Reference format is consistent. | Yes |