# OpenReview forum: "Unsupervised Domain Adaptation for Abdominal Organ Segmentation Using Pseudo Labels and Organ Attention CycleGAN"
_MICCAI.org/2024/Challenge/FLARE — Submitted to FLARE 2024_

### Official Review · Reviewer_vunL · 2025-01-23

**Rating:** 8
**Confidence:** 4

**Review:**

This paper proposes a novel approach leveraging Organ Attention CycleGAN for unsupervised domain adaptation in abdominal organ segmentation, ensuring the preservation of critical anatomical structures during the translation process. Here are my major concerns.

1. Insufficient details regarding the network architecture.

2. In Fig. 3, it would be beneficial to provide an real example for each MRI modality.

---

> ### Author Response · Authors · 2025-03-30
>
> 1. Response: Thank you for the suggestion. We have added more detailed descriptions of the network architecture and training settings in Section 3.2. Specifically, we clarified that the CycleGAN implementation follows the original architecture proposed by Zhu et al. [39], using ResNet-based encoder-decoder generators and PatchGAN discriminators. Changes: Expanded Section 3.2 with architectural and training details for both CycleGAN and nnU-Net V2.
> 2. Response: Thank you for the suggestion. Since our dataset consists of unpaired CT and MRI images, it is not feasible to present real MRI counterparts for the same anatomical slice shown in Fig. 3. As a result, we focused on displaying the translated images generated by our model to highlight structural consistency across modalities. The purpose of Fig. 3 is to demonstrate that the organ structures are preserved during translation into various MRI modalities, rather than to provide a one-to-one visual comparison with real MRI data. Changes: Clarified the purpose of Fig. 3 in the caption, emphasizing structural consistency in translated images under the unpaired setting.

---

### Official Review · Reviewer_Yehu · 2025-01-25

**Rating:** 8
**Confidence:** 5

**Review:**

This paper proposes a novel Organ Attention CycleGAN framework for cross-modality image translation and introduces a prediction consistency algorithm for selecting high-quality pseudo labels during the self-training process.  Here are my major concerns.

    1. The lack of ablation experiments proves the importance of increasing organ attention in CycleGAN and  the selection of consistency-based pseudo labels.
    2. Insufficient details regarding the network architecture.

---

> ### Author Response · Authors · 2025-03-30
>
> 1. Response: Thank you for the insightful comment. Regarding the effectiveness of the consistency-based pseudo labels, we provide a two-stage ablation study in Section 4.1 and Table 5. These results clearly demonstrate improved segmentation performance after incorporating pseudo labels, with average DSC increasing from 68.49% to 76.74% and NSD from 74.22% to 83.19%. For the organ attention mechanism, we designed the CycleGAN to directly integrate attention maps (Section 2.1), and its impact is discussed in the context of reducing anatomical distortion. While we did not isolate this module in a separate ablation due to computational constraints, we will consider adding more detailed component-level analysis in future work. Changes: No changes were made; clarified explanation and references to ablation study in Section 4.1 and design rationale in Section 2.1.
> 2. Response: Thank you for your suggestion. We have updated the manuscript to provide more implementation details. Specifically, Section 3.2 now includes a detailed description of the CycleGAN training setup (e.g., 2D slicing strategy, training epochs, batch size) and the nnU-Net V2 configuration, including encoder type, patch size, learning rate, data augmentation, and optimizer. These additions aim to make the architectural and training design more transparent and reproducible. Changes: Expanded Section 3.2 with architectural and training details for both CycleGAN and nnU-Net V2.

---

### Official Review · Reviewer_fQfE · 2025-01-27
**This work proposed a framework for UDA for organ segmentation based on organ attention cycleGAN and consistency-based pseudo label filtering.**

**Rating:** 6
**Confidence:** 5

**Review:**

1. It would be better to denote what is fake and real in Fig. 1 and keep the model name consistent with the manuscript.
2. Not sure of the effect of organ attention and consistency-based methods
3. If I understand correctly, you have 8 GANs for 8 sequences? So, one of the limitations might be that there are too many models to train.
4. No comparison with other UDA methods

---

> ### Author Response · Authors · 2025-03-30
>
> 1. Response: Thank you for pointing this out. We have revised Figure 1 to clearly denote the "real" and "fake" images involved in the image translation process. Additionally, we ensured that the model names used in the figure are fully consistent with those used throughout the manuscript. Changes: Updated Figure 1 with explicit labels for real and fake images; revised figure caption and ensured naming consistency across Sections 2.1 and 2.2.
>
> 2. Response: We appreciate the reviewer’s interest in the contribution of each module. As presented in Section 4.1 (Table 5), we conducted a two-stage ablation study comparing performance before and after applying consistency-based self-training. This demonstrated clear improvement across most organs. The effectiveness of organ attention is discussed in Section 2.1, where we explain its role in reducing anatomical distortion during CT-to-MRI translation. Together, these modules are shown to meaningfully improve segmentation accuracy. Changes: No major changes; results and descriptions already included in Section 2.1 (Organ Attention) and Section 4.1/Table 5 (Self-training effectiveness).
>
> 3. Response: We thank the reviewer for pointing this out. Our approach trains individual CycleGAN models for each of the 8 MRI modalities, as described in Section 2 and 3.2. This design preserves modality-specific characteristics critical for accurate segmentation. We acknowledge the added training cost and have noted this limitation in the revised “Limitations and Future Work” section, where we also suggest that unified or shared GAN architectures could be explored in future work. Changes: Added a sentence in Section 4.4 acknowledging the scalability limitation and suggesting multi-modal unified GANs as a future direction.
>
> 4. Response: Thank you for the valuable suggestion. Our primary focus was on demonstrating the effectiveness of our proposed method within the FLARE24 evaluation framework. While direct UDA baselines such as AdaptSegNet or SIFA were not included in this version, we clarified this choice in the revised manuscript. We intend to include such comparisons in future extensions of this work. Changes: Added explanation in Section 4.4 regarding the absence of baseline UDA comparisons and discussed future plans to include them.

---

### Decision · Program_Chairs · 2025-03-20

**Decision:**

Accept

**Comment:**

Please adjust CT images window level and width to 400 and 40 respectively.